# Hebbian and Gradient-based Plasticity Enables Robust Memory and Rapid Learning in RNNs

**Yu Duan, Zhongfan Jia, Qian Li, Yi Zhong\*, Kaisheng Ma\***
Tsinghua University, Beijing, China
`{duany19, jzf20}@mails.tsinghua.edu.cn`
`{liqian8, zhongyithu, kaisheng}@tsinghua.edu.cn`

## Abstract

Rapidly learning from ongoing experiences and remembering past events with a flexible memory system are two core capacities of biological intelligence. While the underlying neural mechanisms are not fully understood, various evidence supports that synaptic plasticity plays a critical role in memory formation and fast learning. Inspired by these results, we equip Recurrent Neural Networks (RNNs) with plasticity rules to enable them to adapt their parameters according to ongoing experiences. In addition to the traditional local Hebbian plasticity, we propose a global, gradient-based plasticity rule, which allows the model to evolve towards its self-determined target. Our models show promising results on sequential and associative memory tasks, illustrating their ability to robustly form and retain memories. In the meantime, these models can cope with many challenging few-shot learning problems. Comparing different plasticity rules under the same framework shows that Hebbian plasticity is well-suited for several memory and associative learning tasks; however, it is outperformed by gradient-based plasticity on few-shot regression tasks which require the model to infer the underlying mapping. Code is available at `https://github.com/yuvenduan/PlasticRNNs`.

## 1 Introduction

Biological neural networks can dynamically adjust their synaptic weights when faced with various real-world tasks. The ability of synapses to change their strength over time is called synaptic plasticity, a critical mechanism that underlies animals' memory and learning (Abbott & Regehr, 2004; Stuchlik, 2014; Abraham et al., 2019; Magee & Grienberger, 2020). For example, synaptic plasticity is essential for memory formation and retrieval in the hippocampus (Martin et al., 2000; Neves et al., 2008; Rioult-Pedotti et al., 2000; Kim & Cho, 2017; Nabavi et al., 2014; Nakazawa et al., 2004). Furthermore, recent results show that some forms of synaptic plasticity could be induced within seconds, enabling animals to form memory quickly and do one-shot learning (Bittner et al., 2017; Magee & Grienberger, 2020; Milstein et al., 2021).

To test whether plasticity rules could also aid the memory performance and few-shot learning ability in artificial models, we incorporate plasticity rules into Recurrent Neural Networks (RNNs). These plastic RNNs work like the vanilla ones, except that a learned plasticity rule would update network weights according to ongoing experiences at each time step. Historically, Hebb's rule is a classic model for long-term synaptic plasticity; it states that a synapse is strengthened when there is a positive correlation between the pre- and post-synaptic activity (Hebb, 1949). Several recent papers utilize generalized versions of Hebb's rule and apply it to Artificial Neural Networks (ANNs) in different settings (Miconi et al., 2018; Najarro & Risi, 2020; Limbacher & Legenstein, 2020; Tyulmankov et al., 2022; Rodriguez et al., 2022). With a redesigned framework, we apply RNNs with neuromodulated Hebbian plasticity to a range of memory and few-shot learning tasks. Consistent with the understanding in neuroscience (Magee & Grienberger, 2020; Martin et al., 2000; Neves et al., 2008), we find these plastic RNNs excel in memory and few-shot learning tasks.

---

\*Corresponding authors

Despite being simple and elegant, classical Hebbian plasticity comes with limitations. In multi-layer networks, the lack of feedback signals to previous layers could impede networks' ability to configure their weights in a fine-grained manner and evolve to the desired target (Magee & Grienberger, 2020; Marblestone et al., 2016). In recent years, some authors argue that other forms of plasticity rules in the brain could produce similar effects as the back-propagation algorithm, although the underlying mechanisms are probably different (Sacramento et al., 2018; Whittington & Bogacz, 2019; Roelfsema & Holtmaat, 2018). Inspired by these results, we attempt to model the synaptic plasticity in RNNs as *self-generated* gradient updates: at each time step, the RNN updates its parameters with a self-determined target. Allowing the RNN to generate and evolve to a customized target enables the RNN to configure its weights in a flexible and coordinated fashion. Like Hebb's rule, the proposed gradient-based plasticity rule is task-agnostic. It operates in an *unsupervised* fashion, allowing us to compare these two plasticity rules under the same framework.

In machine learning, learning a plasticity rule is one of the many *meta-learning* approaches (Schmidhuber et al., 1997; Bengio et al., 2013). Although a diverse collection of meta-learning methods have been proposed over the years (Huisman et al., 2021), these meta-learning methods are typically built upon specific assumptions on the task structure (e.g., assume the supervising signals are explicitly given; see Sec. 2 for more detailed discussion). They thus could not be applied to arbitrary learning problems. In contrast, in our networks, the evolving direction of network parameters $d\mathbf{W}/dt$ solely depends on the current network state, i.e., current network parameters and the activity of neurons. Since the designed plasticity rules do not rely on task-specific information (e.g., designated loss function and labels), they could be naturally applied to any learning problems as long as the input is formulated as time series. Therefore, modeling biological plasticity rules also allows us to build more general meta-learners.

Our contribution can be summarized as follows. Based on previous work (Miconi et al., 2019), we formulate a framework that allows us to incorporate different plasticity rules into RNNs. In addition to the local Hebbian plasticity, we propose a novel gradient-based plasticity rule that allows the model to evolve towards self-determined targets. We show that both plasticity rules improve memory performance and enable rapid learning, suggesting that ANNs could benefit from synaptic plasticity similarly to animals. On the other hand, as computational models simulating biological plasticity, our models give insights into the roles of different forms of plasticity in animals' intelligent behaviors. We find that Hebbian plasticity is well-suited for many memory and associative learning tasks. However, the gradient-based plasticity works better in the few-shot regression task, which requires the model to infer the underlying mapping instead of learning direct associations.

## 2 RELATED WORK

**Meta-Learning.** Meta-learning, or "learning to learn", is an evolving field in ML that aims to build models that can learn from their ongoing experiences (Schmidhuber et al., 1997; Bengio et al., 2013). A surprisingly diverse set of meta-learning approaches have been proposed in recent years (Hospedales et al., 2021; Finn et al., 2017; Santoro et al., 2016; Mishra et al., 2018; Lee et al., 2019). In particular, one line of work proposes to meta-learn a learning rule capable of configuring network weights to adapt to different learning problems. This idea could be implemented by training an optimizer for gradient descent (Andrychowicz et al., 2016; Ravi & Larochelle, 2017), training a Hypernetwork that generates the weights of another network (Ha et al., 2017), or meta-learning a plasticity rule which allows RNNs to modify its parameters at each time step (Miconi et al., 2019; Ba et al., 2016; Miconi et al., 2018). Our method belongs to the last category.

Compared to other meta-learning approaches, training plastic RNNs has some unique advantages. Plastic RNNs are general meta-learners that could learn from any sequential input. In contrast, most meta-learning methods cannot deal with arbitrary learning problems due to their assumptions about task formulation. For example, methods that utilize gradient descent in the inner loop (e.g., MAML (Finn et al., 2017), LSTM meta-learner (Ravi & Larochelle, 2017) and GD$^2$ (Andrychowicz et al., 2016)) typically assume that there exist explicit supervising signals (e.g., ground truth) and a loss function that is used to update the base learner. However, such information is often implicit in the real world (e.g., when humans do few-shot learning from natural languages (Brown et al., 2020)). In contrast, plastic RNNs are task-agnostic: they can adapt their weights in an unsupervised manner,

and only a meta-objective is required for meta-training. Besides, the idea of evolving plasticity rules derives from animals, who are still the best meta-learners we have known so far.

Another line of work that is closely related to our gradient-based plasticity rule is meta-learning a loss function. This idea has been applied in reinforcement learning (Houthooft et al., 2018; Oh et al., 2020; Kirsch et al., 2020) and supervised learning (Baik et al., 2021; Bechtle et al., 2021). However, as discussed above, these methods still depend on supervising signals and are thus less general compared to our methods. Moreover, our internal loss generation process is more flexibly determined by ongoing experience. The learning rule in the inner loop is also much more flexible than the usual gradient descent, as each connection has its own learning rate.

**Synaptic Plasticity in ANNs.** Previous work has incorporated synaptic plasticity into ANNs in different settings. For example, Hebbian networks can be explicitly used as storage of associative memories for ANNs (Limbacher & Legenstein, 2020; Schlag et al., 2021b). In addition, Hebb's rule alone can evolve random networks to do simple reinforcement learning tasks (Najarro & Risi, 2020). Miconi et al. (2019; 2018) and Ba et al. (2016) apply generalized Hebbian plasticity to RNNs and find plasticity helpful on tasks including associative learning, pattern memorization, and some simple reinforcement learning tasks. In Differentiable Plasticity (Miconi et al., 2018), the temporal moving average of outer products of pre- and post-synaptic activities is used as the plastic component of network weights. Miconi et al. (2019) extend this method by adding global neuromodulation. A recent paper further extends Hebbian plasticity with short-term dynamics (Rodriguez et al., 2022). Beyond Hebbian plasticity, some recent works explore other ways to capture plasticity in RNNs. For example, some authors use Fast Weight Programmers to update part of the network with a key-value mechanism (Schlag et al., 2021a; Irie et al., 2021). Another line of work on key-value memory networks is also related to synaptic plasticity, where methods including gradient descent (Bartunov et al., 2020; Munkhdalai et al., 2019) and three-factor plasticity rules (Tyulmankov et al., 2021) are proposed to update the memory network.

## 3 METHOD

### 3.1 MODEL FRAMEWORK

Following previous work on Hebbian plasticity (Miconi et al., 2018; 2019; Tyulmankov et al., 2022; Rodriguez et al., 2022), we assume the weights for plastic layers to be the sum of a static part $\tilde{\mathbf{w}}$ and a plastic part $\mathbf{w}$. We initialize the plastic part as $0$ at the beginning of a trial and update the plastic part throughout the trial. We adopt a general architecture of RNNs as shown in Figure 1 (left). Both the RNN and the last linear layer are plastic. The encoder is a plastic linear layer in most tasks; the only exception is the one-shot image classification task, in which case the encoder is a non-plastic Convolutional Neural Network (CNN). The model output $\mathbf{o}_t$ is the concatenation of three parts: a scalar $\tilde{\eta}_t$, which modulates global plasticity by controlling how fast parameters change; a vector $\mathbf{y}_t$ representing the model prediction; and an additional vector $\tilde{\mathbf{y}}_t$ which allows more flexible control of weights for networks with gradient-based plasticity, as described in more detail in Sec. 3.3.

We summarize the general framework for training plastic RNNs in Algorithm 1. In the inner loop, i.e., each time step of RNN, the network learns from ongoing experiences and adjusts its weights accordingly. In the outer loop, network parameters, including those that define the learning rules in the inner loop, are meta-trained with gradient descent. Conceptually, the outer loop corresponds to the natural evolution process where the biological synaptic plasticity is evolved. In our framework, the network updates its parameters in an unsupervised fashion, i.e., the computation of $\Delta\mathbf{w}$ in Algorithm 1 does not depend on the ground truth $\tilde{\mathbf{y}}_t$. The network must thus learn to adapt its parameters given only the input $\mathbf{x}_t$. In some of our tasks, ground truth is given as part of the input for the model to learn the association between observations and targets (see Sec. 4). We choose not to use any external supervising signals to follow the tradition of Hebbian plasticity, which does not depend on explicit supervising signals. Our formulation is thus a more realistic setting where the model must learn to identify the supervising signals from the input.

### 3.2 HEBBIAN PLASTICITY

We first discuss Hebbian plasticity with global neuromodulation. Recall that we assume the weight in a plastic layer $l$ to be the sum of a static part $\tilde{\mathbf{w}}_l$ and a plastic part $\mathbf{w}_l$. The plastic component

---

**Algorithm 1** Training algorithm of plastic RNNs

---

**Require:** Trial Distribution $\mathcal{T}$, loss function $\ell$
**Require:** An RNN $f$ and its initial parameters $\tilde{\mathbf{w}}$, connection-specific learning rates $\boldsymbol{\alpha}$
 1: **while** not converged **do (Outer Loop)**
 2:     Sample a batch of trials $\mathcal{T}_i \sim \mathcal{T}$
 3:     Initialize meta-loss $L = 0$
 4:     **for** each trial $\mathcal{T}_i = (\mathbf{x}, \mathbf{y}')$ **do**
 5:         Initialize the plastic weights $\mathbf{w}_0 \leftarrow \mathbf{0}$, initialize the hidden state $\mathbf{h}_0 \leftarrow \mathbf{0}$
 6:         **for** time step $t = 1, 2, ..., T$ **do (Inner Loop)**
 7:             Get model prediction and new hidden state $(\mathbf{y}_t, \mathbf{h}_t) \leftarrow f(\mathbf{x}_t, \mathbf{h}_{t-1}; \mathbf{w}_{t-1} + \tilde{\mathbf{w}})$
 8:             Compute the internal learning rate $\eta(t)$ according to equation 2
 9:             Update the plastic parameters $\mathbf{w}_t \leftarrow (1 - \eta(t))\mathbf{w}_{t-1} + \eta(t)\boldsymbol{\alpha} \circ \Delta\mathbf{w}$, where $\Delta\mathbf{w}$ is
    computed with Hebbian or the gradient-based rule, see Sec. 3.2 and 3.3 for more details
10:             Accumulate meta-loss $L \leftarrow L + \ell(\mathbf{y}_t, \mathbf{y}'_t)$
11:         **end for**
12:     **end for**
13:     Perform gradient descent w.r.t. $L$ to update $\tilde{\mathbf{w}}, \boldsymbol{\alpha}$
14: **end while**

---

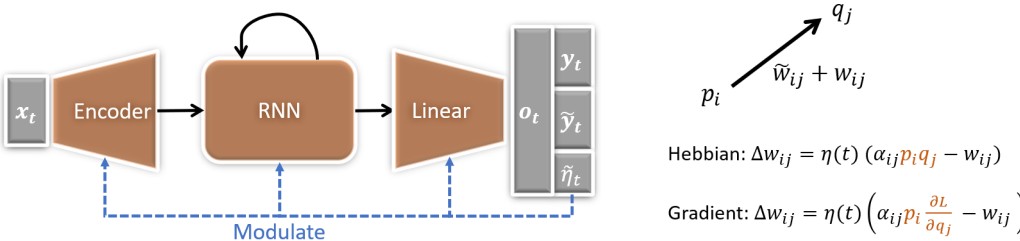

Figure 1: **Left:** Model Architecture (see Sec. 3.1). **Right:** Comparison of different learning rules in a linear layer without nonlinearity. The difference between the two learning rules is highlighted. See Sec. 3.2 and 3.3 for more details.

$\mathbf{w}_l(t)$ is updated at each time step according to the outer product of pre-synaptic activity $\mathbf{p}_l(t)$ and post-synaptic activity $\mathbf{q}_l(t)$:

$$
\begin{aligned}
\mathbf{q}_l(t) &= \sigma_l \left( \mathbf{b}_l + (\mathbf{w}_l(t) + \tilde{\mathbf{w}}_l)^T \mathbf{p}_l(t) \right), \\
\mathbf{w}_l(t+1) &= (1 - \eta(t))\, \mathbf{w}_l(t) + \eta(t)\boldsymbol{\alpha}_l \circ (\mathbf{p}_l(t)\mathbf{q}_l^T(t)), \quad \mathbf{w}_l(0) = \mathbf{0},
\end{aligned}
\tag{1}
$$

where $\sigma_l$ is the activation function, $\circ$ denotes element-wise product, and $\boldsymbol{\alpha}_l$ are learnable parameters that are initialized from $\mathcal{U}[-1, 1]$. $\boldsymbol{\alpha}_l$ acts as connection-specific learning rates that allow each synapse to have different learning rules (e.g., Hebbian or anti-Hebbian). Previous work on Hebbian plasticity has shown the benefit of having connection-specific plasticity over homogeneous plasticity (Miconi et al., 2018), and we found the same results in our experiments. The decay term, which is similar to the weight decay used in gradient descent algorithms, has been introduced in some recent Hebbian models (Miconi et al., 2018; Tyulmankov et al., 2022) to prevent the weight from exploding. $\eta(t)$ is the *internal learning rate* that controls the global plasticity, calculated as follows:

$$
\begin{aligned}
\eta(t) &= \eta_0 \times \text{Sigmoid}(\tilde{\eta}_t) \times \min \left\{ 1, \frac{\text{max\_norm}}{\|\boldsymbol{\delta}_t\|_2} \right\}, \quad \text{where} \\
\boldsymbol{\delta}_t &= \text{Concat} \left( \text{Vec}(\mathbf{p}_l(t)\mathbf{q}_l^T(t)) | l \in S \right),
\end{aligned}
\tag{2}
$$

$\eta_0$ is a hyperparameter that controls the maximal learning rate, $\text{Vec}(\cdot)$ denotes the vectorization of a matrix, $\text{Concat}(\cdot)$ denotes the concatenation of a collection of vectors, and $S$ is the set of plastic layers in the network. We scale the internal learning rate according to the norm of $\boldsymbol{\delta}_t$ to prevent weights from changing too quickly. We use $\text{max\_norm} = 1$ and $\eta_0 = 0.2$ in all experiments.

Controlling the global plasticity with a self-generated signal $\eta(t)$ is well-motivated from the biological perspective. In animals, neurotransmitters, especially dopamine, play an essential role in

modulating plasticity and consequently influence animals' memory and learning (Cohn et al., 2015; Katiuska et al., 2009; Kreitzer & Malenka, 2008; Nadim & Bucher, 2014). Theoretical models usually model neuromodulation as a global factor due to the volume transmission of neurotransmitters (Magee & Grienberger, 2020). Allowing the brain to adaptively modulate synaptic plasticity enables reward-based learning (Gu, 2002; Pignatelli & Bonci, 2015) and active control of forgetting (Berry et al., 2012; 2015). Previous work has shown the benefit of such adaptive learning rates on Hebbian plasticity (Miconi et al., 2019). In our experiments, we empirically demonstrate that neuromodulation is helpful for both Hebbian and gradient-based plasticity, validating the biological understandings from a computational perspective.

## 3.3 GRADIENT-BASED PLASTICITY

For RNNs with gradient-based plasticity, at each time step $t$, we first calculate the *internal loss* on the model output $\mathbf{o}_t$:

$$L(t) = \frac{1}{\dim(\mathbf{o}_t)}\|\mathbf{w}_{\text{out}}^T \mathbf{o}_t\|_2^2 = \frac{1}{\dim(\mathbf{o}_t)}\|\mathbf{w}_{\text{out}}^T \text{Concat}(\mathbf{y}_t, \tilde{\mathbf{y}}_t, \eta_t)\|_2^2, \qquad (3)$$

where $\mathbf{w}_{\text{out}}$ are parameters initialized as 1 that are trained in the outer loop. All three components of the model output are used to calculate the internal loss. $\tilde{\mathbf{y}}_t$ enables the model to meta-learn a customized internal loss function that does not only depend on model prediction. In practice, we find a four-dimensional $\tilde{y}_t$ works well enough. Note that $\tilde{\mathbf{y}}_t$ does not affect network dynamics in Hebbian plasticity. The internal loss term does not involve the ground truth; it can thus be viewed as a self-generated target that the model wants to optimize. We then update the plastic parameters as follows:

$$\mathbf{w}_l(t+1) = (1 - \eta(t))\mathbf{w}_l(t) + \eta(t)\boldsymbol{\alpha}_l \circ \frac{\partial L(t)}{\partial \mathbf{w}_l(t)}, \mathbf{w}_l(0) = \mathbf{0};$$

$$\mathbf{b}_l(t+1) = (1 - \eta(t))\mathbf{b}_l(t) + \eta(t)\boldsymbol{\beta}_l \circ \frac{\partial L(t)}{\partial \mathbf{b}_l(t)}, \mathbf{b}_l(0) = \mathbf{0}; \qquad (4)$$

$$\boldsymbol{\delta}_t = \text{Concat}\left(\text{Vec}\left(\frac{\partial L(t)}{\partial \mathbf{w}_l(t)}\right), \frac{\partial L(t)}{\partial \mathbf{b}_l(t)}\middle| l \in S\right),$$

where $\boldsymbol{\beta}$ are learnable element-wise learning rates just like $\boldsymbol{\alpha}$; $\eta(t)$ is defined the same way as in equation 2, except that $\boldsymbol{\delta}_t$ now denotes the concatenation of *gradients* of all plastic parameters. One difference with Hebbian plasticity is that the bias terms $\mathbf{b}_l$ are also plastic and updated similarly to weights. The gradient-based plasticity rule resembles the usual gradient descent, but the connection-specific learning rates $\boldsymbol{\alpha}$ and $\boldsymbol{\beta}$ allow additional flexibility.

Figure 1 (right) shows a conceptual comparison between Hebbian and gradient-based plasticity. The main difference is that, for gradient-based plasticity, the gradient of the post-synaptic activity replaces the activity itself, thus allowing update signals to propagate to previous layers. Two plasticity rules are particularly similar in the last linear layer, where the gradient-based plasticity rule takes the same form as Hebb's rule up to a constant scaling vector (Sec. A.1). In other words, the last linear layer will still follow Hebb's rule in a network with the proposed gradient-based plasticity.

## 4 EXPERIMENTS

Inspired by the hypotheses on the role of synaptic plasticity in animals (Magee & Grienberger, 2020; Martin et al., 2000; Neves et al., 2008), we conduct experiments to test the following hypotheses:

1. Plasticity helps to form and retain memory and enlarges the memory capacity. In our network, we expect the plastic weights to act as extra memory storage in addition to the hidden states of RNN. We use the copying task and the cue-reward association task to test memory capability.

2. Plasticity helps the model to learn rapidly from their experiences and observations. We test this hypothesis with one-shot image classification and few-shot regression tasks.

To make fair comparisons between models, we scale the size of hidden layers of different models so that all models have approximately the same number of parameters. We use four different random seeds and average the results in all experiments. See Sec. A.2 for more implementation details.

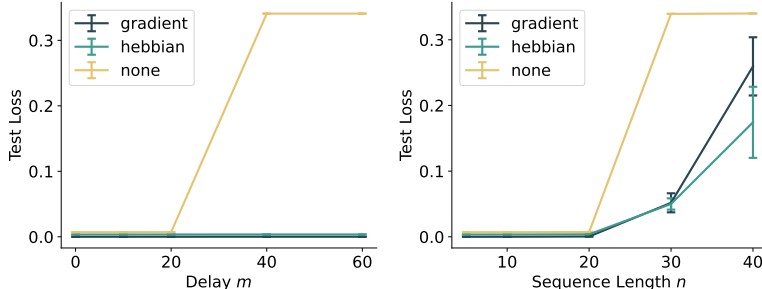

Figure 2: Performance of LSTM models with different plasticity rules on the copying task. Error bars represent the SEM of four random runs. **Left:** Performance with different $m$ when $n = 5$. **Right:** Performance with different $n$ when $m = 0$.

## 4.1 COPYING TASK

We first test our models on a sequential copying task. In each trial, we generate a random sequence of length $n$. After a delay of $m$ steps, the model must reproduce the sequence in its original order. The total length of one trial is thus $2n + m$. We calculate the MSE loss as the criterion for model performance. For more details, see Sec. A.3.1.

We conduct two sets of experiments to compare our models with non-plastic baseline models. First, to test the ability of the models to retain memory during a delay, we set $n = 5$ and vary the number of delay steps $m$. Second, to measure the models' sequential memory capacity, we set $m = 0$ and vary the length of the sequence to be remembered. The results of LSTM models are shown in Figure 2 (see Figure 6 for results of RNNs). Indeed, we find plastic models exhibit larger memory capacity and are able to remember the sequence after a long delay. In contrast, baseline models are typically stuck on chance performance when the delay is large (see Figure 7 for learning curves). The qualitative difference between plastic and non-plastic models shows the plastic RNNs' ability to capture long-term dependency when meta-trained with gradient descent.

## 4.2 CUE-REWARD ASSOCIATION

Associative memory refers to the ability to remember the relationship between unrelated items. For animals, the dependency of associative memory on synaptic plasticity is well-documented in neuroscience literature (Morris et al., 1986; Kim & Cho, 2017; Nakazawa et al., 2004). To evaluate if plasticity also helps the formation of associative memory in artificial RNNs, we train our models to quickly associate cues with corresponding rewards. In each trial, we first sample $n$ random cues and their corresponding rewards. We randomly choose a cue at each time step and present it to the model. The model is expected to answer the corresponding reward. During the first half of the trial, the ground truth is also given in input for the model to learn the association. Please refer to Sec. A.3.2 for more details.

The result is shown in Figure 3. Plastic models quickly converge to reasonable solutions with both the RNN and LSTM backbone. The two plasticity rules perform similarly, but the gradient-based plasticity appears more compatible with the LSTM backbone.

## 4.3 ONE-SHOT IMAGE CLASSIFICATION

To test models' ability of rapid learning, we train our models on the one-shot image classification task, a classic benchmark used in meta-learning. Here we consider the sequential version of 5-way one-shot image classification on MiniImageNet (Vinyals et al., 2016) and CIFAR-FS (Bertinetto et al., 2019). Similar to the previous task, in each trial, the model needs to learn the association between image embedding and the corresponding class in the training stage, then infer the class of novel images in the testing stage. Following previous work, we choose a regular 4-layer CNN or ResNet-12 architecture as the image encoder (Lee et al., 2019). We describe more task details in Sec. A.3.3.

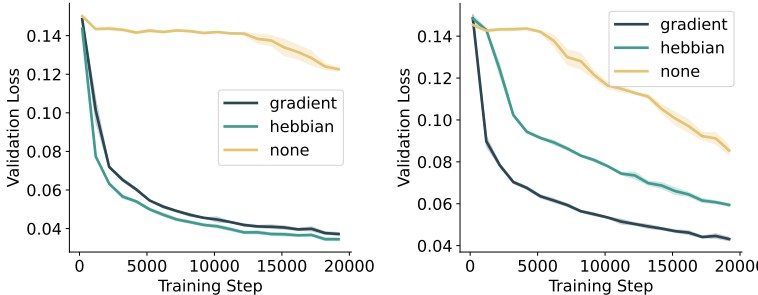

Figure 3: Validation loss curves in the cue-reward association task. The shaded area represents SEM on four random runs. Here $n = 5$ and trial length = 20. **Left:** RNN models. **Right:** LSTM models.

| Models | Conv-4 | | ResNet-12 | |
| --- | --- | --- | --- | --- |
| | **CIFAR-FS** | **miniImageNet** | **CIFAR-FS** | **miniImageNet** |
| LSTM, Non-Plastic | $49.9 \pm 0.5$ | $46.8 \pm 0.7$ | $52.0 \pm 0.9$ | $20.3 \pm 0.3$ |
| LSTM, Hebbian | $50.0 \pm 0.7$ | $46.6 \pm 0.8$ | $49.3 \pm 1.3$ | $33.2 \pm 7.5$ |
| LSTM, Gradient | $50.5 \pm 0.6$ | $47.0 \pm 0.3$ | $50.6 \pm 1.1$ | $28.1 \pm 10.4$ |
| RNN, Non-Plastic | $39.9 \pm 0.8$ | $44.1 \pm 0.7$ | $41.5 \pm 2.4$ | $42.3 \pm 0.6$ |
| RNN, Hebbian | $\mathbf{55.5 \pm 1.0}$ | $\mathbf{49.8 \pm 0.5}$ | $\mathbf{59.6 \pm 1.5}$ | $\mathbf{50.4 \pm 1.1}$ |
| RNN, Gradient | $51.2 \pm 2.6$ | $47.9 \pm 1.2$ | $52.8 \pm 4.4$ | $\mathbf{50.5 \pm 0.4}$ |
| MAML (Finn et al., 2017) | $58.9 \pm 1.9$ | $48.7 \pm 1.8$ | - | - |
| ProtoNet (Snell et al., 2017) | $55.5 \pm 0.7$ | $53.5 \pm 0.6$ | $72.2 \pm 0.7$ | $59.3 \pm 0.6$ |
| COSOC (Luo et al., 2021) | - | - | - | $69.3 \pm 0.5$ |

Table 1: Model performance (test accuracy) on the one-shot image classification task compared to other methods. 95% confidence interval is shown. Data for ProtoNet is from (Lee et al., 2019).

The test performance of our models is reported in Table 1. Both plasticity rules improve the performance of RNN models by a large margin, with the Hebbian plasticity performing slightly better. The learning curves (Figure 12) show that non-plastic RNNs can also overfit the training set, but the generalization gap is much larger than the plastic RNNs. These observations suggest that plasticity not only increases representation power but also provides a powerful inductive bias that inherently strengthens models' ability to learn from their environments quickly. Interestingly, even though the non-plastic LSTM consistently outperforms the non-plastic RNN, this is no longer the case if plasticity is introduced. We infer that plastic weights provide stable memory storage like the cell states in LSTMs. As a result, the original advantages of LSTM models might no longer exist.

Plastic networks have comparable performance to other meta-learning methods when we limit the visual encoder to be a 4-layer CNN. However, we did not find plastic networks significantly benefit from a deep vision encoder like the recent work on few-shot image classification (Lee et al., 2019; Huisman et al., 2021). In recent years, methods that get the best results on few-shot learning benchmarks (e.g., MetaOptNet (Lee et al., 2019), COSOC (Luo et al., 2021)) are also exclusively designed for few-shot image classification. Unlike our plastic RNNs, these methods are difficult to apply to other learning problems or memory tasks. Instead of striving to get the best performance on a specific task, our goal is to build a general architecture that tackles a wide range of memory and learning problems.

### 4.4 FEW-SHOT REGRESSION

We test our models on a regression task to further evaluate the performance of few-shot learning. In each trial, we randomly generate a mapping $f : [-1, 1]^d \to \mathbb{R}$, which is either a linear function or a small MLP. The model needs to learn the underlying mapping from $K$ observations, i.e, $K$ pairs of $(\mathbf{x}_t, f(\mathbf{x}_t))$, then make predictions on $f(\mathbf{x}_t)$ given $\mathbf{x}_t$. In our experiments, we use $K = 10$ or 20. In the case of $K = 10$, $K$ is even smaller than the free parameters in $f$, making the task more challenging. More details of the task are described in Sec. A.3.4.

| Models | Linear | | MLP | |
|---|---|---|---|---|
| | $K = 10$ | $K = 20$ | $K = 10$ | $K = 20$ |
| LSTM, Non-Plastic | $.605 \pm .002$ | $.402 \pm .010$ | $.241 \pm .043$ | $.087 \pm .006$ |
| LSTM, Hebbian | $.446 \pm .003$ | $.229 \pm .001$ | $.282 \pm .013$ | $.112 \pm .002$ |
| LSTM, Gradient | $\mathbf{.300 \pm .002}$ | $\mathbf{.107 \pm .001}$ | $\mathbf{.142 \pm .001}$ | $\mathbf{.036 \pm .001}$ |
| RNN, Non-Plastic | $.605 \pm .002$ | $.469 \pm .001$ | $.465 \pm .015$ | $.383 \pm .007$ |
| RNN, Hebbian | $.378 \pm .059$ | $.200 \pm .042$ | $.165 \pm .001$ | $.077 \pm .028$ |
| RNN, Gradient | $\mathbf{.301 \pm .001}$ | $\mathbf{.108 \pm .001}$ | $\mathbf{.142 \pm .001}$ | $\mathbf{.036 \pm .001}$ |

Table 2: Test error on the $K$-shot regression task. 95% confidence interval is shown.

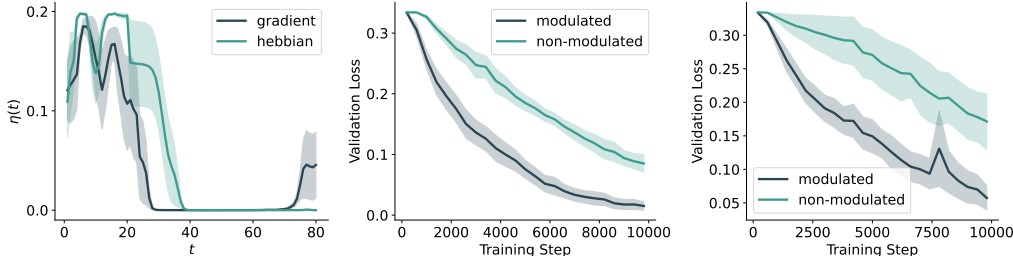

Figure 4: **Left:** Dynamics of $\eta(t)$ in the copying task, we average the result across the 6400 trials in the test set. The shaded area reflects the SEM of models with different random seeds. LSTM models are shown. The sequence length $n = 20$ and delay $m = 40$. **Middle and Right:** Learning curves of different models in the copying task. Here we compare using an adaptive internal learning rate ("modulated") and using a fixed one ("non-modulated"). **Middle:** models with Hebbian plasticity. **Right:** models with gradient-based plasticity.

Model performance is shown in Table 2. Unlike previous tasks, the proposed gradient-based plasticity consistently produces the best result, although Hebbian plasticity also improves the performance in most cases. We infer that such a difference is caused by the need for inference in the few-shot regression task. In this task, the model not only needs to remember the observations but also infer the underlying mapping $f$ from these observations, which necessitates complex calculations not required in tasks such as associative learning. Models with gradient-based plasticity are more capable of learning the underlying rules, probably because they can leverage back-propagation to optimize their circuits over the desired target. In contrast, models with Hebbian plasticity are relatively weak at evolving their network weights to reach any given target. A local learning rule like Hebb's rule might be good enough for learning direct associations. However, the lack of feedback signals to prior layers makes it hard for the whole network to evolve in a coordinated fashion.

## 4.5 ANALYSIS AND ABLATION STUDY

Recall that we use the internal learning rate $\eta(t)$ in our plastic networks to model biological neuromodulation. $\eta(t)$ is a key quantity that controls how much information is stored and discarded in plastic weights at each time step $t$. By tuning $\eta(t)$ in an adaptive way, the network can effectively choose to learn from experience quickly or retain the previously learned knowledge. Indeed, as shown in Figure 4, such adaptive behavior of $\eta(t)$ is observed in our experiments. In the copying task, the network learns to set large $\eta(t)$ when the sequence is presented to the model. $\eta(t)$ quickly decays to 0 during the delay, probably because the model learns to preserve the memory stored in plastic weights. In terms of task performance, we find an adaptive $\eta(t)$, i.e., instead of setting $\eta(t)$ to a fixed value, consistently leads to improvements for both plasticity rules. Similar results are also observed in other tasks (Figure 10). These results suggest that neuromodulation is the key to a flexible and robust plasticity-based memory system.

We conduct additional ablation studies on the cue-reward association task; the detailed figures are shown in Sec. A.4. We find plastic RNNs consistently outperform non-plastic ones when larger or smaller learning rates are used (Figure 8), and our default learning rate is appropriate for plastic and non-plastic models. We find the maximal learning rate $\eta_0$ to be a key hyperparameter; we

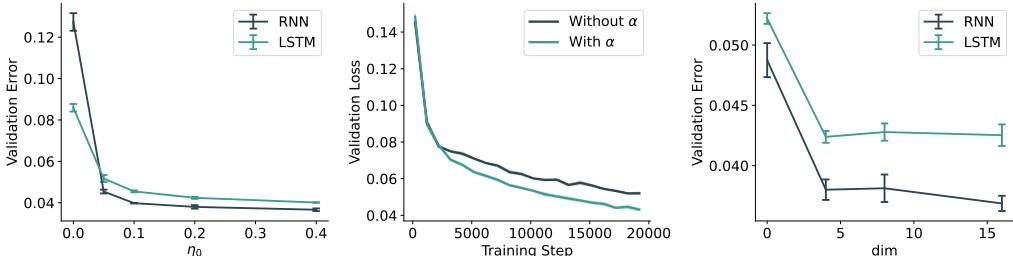

Figure 5: Ablation study on the cue-reward association task, gradient-based plasticity is used for all three panels. **Left:** Effect of $\eta_0$ on final validation error, note that $\eta_0 = 0$ means no plasticity. **Middle:** Effect of connection-specific learning rates $\boldsymbol{\alpha}$ on validation loss curve. **Right:** Effect of $\dim(\tilde{\mathbf{y}}_t)$ on final validation error.

set $\eta_0 = 0.2$ for our main experiments to balance model capability and training stability (Figure 5 left, also see Figure 9). We show that randomly-initialized connection-specific learning rates $\boldsymbol{\alpha}$ consistently work better than a fixed global learning rate (Figure 5 middle, also see Figure 11). For the gradient-based plasticity, we find that using $\tilde{\mathbf{y}}_t$ improves performance, but $\dim(\tilde{\mathbf{y}}_t) = 4$ works good enough (Figure 5 right). In addition, on the copying task, when $n = 5$ and $m = 40$, we find that setting max_norm $= 100$ instead of 1 causes gradient explosion (so that the training loss becomes nan) in 3 out of 4 random runs, illustrating the necessity of tuning down $\eta(t)$ when the norm of change $\|\boldsymbol{\delta}_t\|$ exceeds an appropriate threshold (equation 2).

## 5 DISCUSSION

In this work, we draw inspiration from biological synaptic plasticity and propose to incorporate different forms of plasticity into RNNs. We highlight the advantage of neuromodulated plasticity by comparing plastic RNNs against non-plastic ones on a range of challenging memory and few-shot learning tasks. In resonance with hypotheses from neuroscience (Magee & Grienberger, 2020; Martin et al., 2000; Neves et al., 2008), we show that adopting plasticity in RNNs improves their memory performance and helps them to learn from observations quickly. Moreover, we go beyond the traditional Hebbian plasticity and design a novel gradient-based plasticity where the model can flexibly adapt its weights with a self-generated target. Our experiments illustrate the feasibility of training RNNs capable of doing gradient updates where both the learning rule and the internal loss function are meta-trained.

By comparing two plasticity rules under the same framework, we find both of them have pros and cons. The classical Hebbian plasticity, which is computationally more efficient, proved sufficient for robust memory storage and enabled the network to do simple forms of learning. However, results on the few-shot regression task show an example of how networks could benefit from non-local learning rules where error signals are propagated to prior layers. Just like different forms of plasticity rules have been found in different brain regions accountable for different cognitive functions (Magee & Grienberger, 2020), we believe that different plasticity rules are suitable for different tasks in ANNs. For neuroscientists, our work also provides a computational framework that can potentially offer insights into the functions of different plasticity rules in the brain, which is still challenging to directly test in animal experiments (Neves et al., 2008; Magee & Grienberger, 2020).

Despite the promising results, training plastic RNNs with the current deep learning paradigm comes with challenges. Plastic weights are intrinsically unstable, and methods that stabilize the model (e.g., clip weights, normalization) could also limit the model's capability. In addition, training plastic RNNs with back-propagation requires the plastic weight at each step to be stored in the computational graph, causing extensive memory usage on GPUs. As a result, applying plastic RNNs in more challenging settings (e.g., large-scale language modeling) takes more engineering effort. We hope our work can encourage future researchers to further improve the engineering framework, explore other designs of plasticity rules, and investigate the benefit of synaptic plasticity in an even more comprehensive range of tasks.

ACKNOWLEDGEMENTS

We are thankful for Liyuan Wang and Yudi Xie for their helpful feedback. This work was supported by a NSF of China Project (Research on the power supply of implanted neural dust clusters at the sub-neural cell scale, 041302027).

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

## A  APPENDIX

### A.1  COMPARISON OF TWO PLASTICITY RULES

Here we show that for the proposed gradient-based plasticity, the dynamic of the parameters in the last linear layer is Hebbian.

$$
\begin{aligned}
\mathbf{w}_l(t+1) - \mathbf{w}_l(t) &= \eta(t)\left(\boldsymbol{\alpha}_l \circ \frac{\partial L(t)}{\partial \mathbf{w}(t)} - \mathbf{w}_l(t)\right) \\
&= \eta(t)\left(\boldsymbol{\alpha}_l \circ \left(\mathbf{p}_l(t)\frac{\partial L(t)}{\partial \mathbf{q}_l^T(t)}\right) - \mathbf{w}_l(t)\right) \\
&= \eta(t)\left(\boldsymbol{\alpha}_l \circ \left(\mathbf{p}_l(t)\left(\mathbf{q}_l(t) \circ \frac{2\mathbf{w}_{\text{out}} \circ \mathbf{w}_{\text{out}}}{\dim(\mathbf{o}_t)}\right)^T\right) - \mathbf{w}_l(t)\right) \\
&= \eta(t)\left(\boldsymbol{\alpha}_l \circ \left(\mathbf{p}_l(t)\left(\mathbf{q}_l(t) \circ \mathbf{v}\right)^T\right) - \mathbf{w}_l(t)\right),
\end{aligned}
\tag{5}
$$

where $\mathbf{v}$ is a scaling vector that does not change in a trial. In comparison, with Hebbian plasticity, we have

$$
\mathbf{w}_l(t+1) - \mathbf{w}_l(t) = \eta(t)\left(\boldsymbol{\alpha}_l \circ \left(\mathbf{p}_l(t)\mathbf{q}_l^T(t)\right) - \mathbf{w}_l(t)\right).
\tag{6}
$$

Please refer to Sec. 3 for the meaning of notations.

## A.2 Implementation Details

### A.2.1 Architecture

For all tasks, the RNN and the last linear layer are plastic. For the one-shot image classification task, the encoder is a CNN followed by a linear layer, both the CNN and the following linear layer are non-plastic. For other tasks, the encoder is a plastic linear layer. In all experiments, the size of the RNN input layer is equal to the hidden size of the RNN. We apply the ReLU activation after the linear layer in the encoder. We use ReLU activation in vanilla RNNs; we use the customary activation functions in LSTMs. Layer weights and biases are initialized between $[-1/N, 1/N]$, where $N$ is the number of output neurons, i.e., layers are initialized in a fan-out fashion. However, in the one-shot image classification task, considering the large dimension of the CNN embedding (1600 for Conv-4 and 16000 for ResNet-12), the non-plastic linear layer following the CNN is initialized in a fan-in fashion for better performance.

Due to the extra parameter of synapse-wise learning rate $\boldsymbol{\alpha}$, the number of parameters in plastic models almost double. We thus scale the hidden size of the baseline model by $1.5\times$. We also scale the hidden size of RNN models by $2\times$. See Table 3 for details.

| Models | Hidden size | Hidden size in One-shot Image Classification |
|---|---|---|
| LSTM, Non-Plastic | 192 | 384 |
| LSTM, Hebbian | 128 | 256 |
| LSTM, Gradient | 128 | 256 |
| RNN, Non-Plastic | 384 | 768 |
| RNN, Hebbian | 256 | 512 |
| RNN, Gradient | 256 | 512 |

Table 3: Model size in different tasks. We scale the hidden size so that all models have a similar number of parameters.

### A.2.2 Training

| Tasks | Training Steps | Validation Interval |
|---|---|---|
| Copying | 10000 | 200 |
| Cue-reward association | 20000 | 200 |
| One-shot image classification | 40000 | 1000 |
| Few-shot regression | 10000 | 200 |

Table 4: Total number of training steps and the validation interval (i.e., how often do we perform a validation) in different tasks.

For meta-training, we use the AdamW optimizer (Loshchilov & Hutter, 2017) with a learning rate of $10^{-3}$. We use batch size $= 64$ in all experiments. Gradients are clipped at norm $5.0$. In the one-shot image classification experiments, we use a weight decay of $5 \times 10^{-4}$; we use the cosine annealing scheduler with the final learning rate set to $10^{-4}$. In other tasks, we set weight decay to zero since we assume infinite training samples; learning rate schedulers are not used. Both the validation and the test set comprise 6400 trials (i.e., 100 batches). The number of training steps and the validation interval in different tasks is shown in Table 4. We test the model with the best validation error (or accuracy) on the test set and then report the test error (or accuracy).

Note that to meta-train plastic RNNs with gradient-based plasticity, we need to compute gradient-of-gradients, which makes the training of gradient-based plasticity about 20% slower than Hebbian plasticity on the cue-reward association task.

## A.3 TASK DETAILS

### A.3.1 COPYING TASK

In each trial, we generate a sequence of random numbers $r_i$ drawn from $\mathcal{U}[-1, 1]$ of length $n$. Recall that the model must reproduce the sequence after a delay of $m$ steps. At each time step $t$, the input is a three dimensional vector $\mathbf{x}_t$, where:

- If $t \in [1, n]$, $\mathbf{x}_t = (r_t, 1, 0)$. In other words, we present the sequence to the model during the first $n$ steps;
- If $t \in [n+1, n+m]$, $\mathbf{x}_t = (0, 0, 0)$;
- If $t \in [n+m+1, 2n+m]$, $\mathbf{x}_t = (0, 0, 1)$. So the last entry is a binary variable indicating whether $t \geq m + n$.

We calculate the MSE loss as the criterion for model performance, i.e.,

$$L = \frac{1}{n} \sum_{t=1}^{n} (r_t - y_{n+m+t})^2,$$

where $y_t$ denotes the model prediction. The same loss is used for meta-training.

### A.3.2 CUE-REWARD ASSOCIATION

Let the trial length be $m$. Recall that we use $m = 20$ in our experiments. In each trial, we first sample $n$ random cues $\mathbf{c}_1, ..., \mathbf{c}_n$ from $[0, 1]^d$ where $d = 14$. We also sample the corresponding rewards $r_1, ..., r_n$ from $[0, 1]$. At each time step $t$, we randomly sample $i_t \in \{1, 2, ..., n\}$ and a small Gaussian noise $\boldsymbol{\delta}_t$ with $\sigma = 0.1$. The input is a vector of size $d + 2$, $\mathbf{x}_t = \text{Concat}(\mathbf{c}_{i_t} + \boldsymbol{\delta_t}, \tilde{r}_t, b_t)$, where:

- If $t \in [1, m/2]$, $\tilde{r}_t = r_{i_t}, b_t = 0$.
- If $t \in [m/2 + 1, m]$, $\tilde{r}_t = 0, b_t = 1$.

We calculate the MSE loss over the whole trial as the criterion for model performance, i.e.,

$$L = \frac{1}{m} \sum_{t=1}^{m} (r_{i_t} - y_t)^2.$$

The same loss is used for meta-training.

### A.3.3 ONE-SHOT IMAGE CLASSIFICATION

We first give more details on the dataset. Previous work reported that Hebbian plasticity improves one-shot image classification performance on Omniglot dataset (Miconi et al., 2018). However, Omniglot is not a challenging image dataset and models can easily get near-perfect performance (Mishra et al., 2018; Miconi et al., 2018); therefore, recent works tend to choose MiniImageNet (Vinyals et al., 2016) and CIFAR-FS (Bertinetto et al., 2019) instead. The miniImageNet dataset consists of 100 image classes, and there are 600 images of size $84 \times 84$ for each class. The CIFAR-FS dataset consists of the 100 image classes from CIFAR-100 (Krizhevsky et al., 2009); each class has 600 images of size $32 \times 32$. For both datasets, the 100 image classes are split into 64, 16, and 20 classes for meta-training, meta-validation, and meta-testing, respectively. We use the customary way to partition the datasets (Ravi & Larochelle, 2017; Lee et al., 2019).

Now we describe the details of the sequential version of 5-way 1-shot image classification. In each trial, we randomly sample five image classes from the dataset. Unlike previous tasks, the input in this task is an image-label pair $(\mathbf{a}_t, \mathbf{b}_t)$, where $\mathbf{a}_t$ is a random image from one of the five chosen classes, $\mathbf{b}_t$ is a 5-dimensional vector.

- For $1 \leq t \leq 5$, (the training stage), $\mathbf{b}_t$ is the one-hot encoding of the label $\tilde{y}_t$, where $\tilde{y}_t \in \{1, 2, 3, 4, 5\}$.

- For $6 \leq t \leq 10$, (the testing stage), $\mathbf{b}_t = \mathbf{0}$, the model needs to predict the label with learned associations.

In both the training and the testing stage, the model will see precisely one image from each chosen class, and the order of labels is randomized. During meta-training, we average the cross entropy loss on the whole testing stage, i.e.,

$$L = \frac{1}{5} \sum_{t=6}^{10} \text{CrossEntropy}(\mathbf{y}_t, \tilde{y}_t).$$

Since the model might utilize the prior knowledge that exactly one image from each class is presented in the testing stage, we only consider the first test image when we calculate the test accuracy.

In order for the model to process the paired input $(\mathbf{a}_t, \mathbf{b}_t)$, we first calculate the image embedding of $\mathbf{a}_t$, the embedding is then projected to a lower dimension by a non-plastic linear layer and concatenated with $\mathbf{b}_t$. The dimension of the resulting vector is the same as the hidden size of the RNN. We adopt some common techniques to alleviate meta-overfitting, including data augmentations and dropblock (Ghiasi et al., 2018) in ResNet models. We use the same data augmentations and dropblock configurations used in MetaOptNet (Lee et al., 2019).

### A.3.4 FEW-SHOT REGRESSION

Recall that in each trial, we first generate a mapping $f : [-1, 1]^d \to \mathbb{R}$. At each time step $t$, we sample a random vector $\mathbf{v}_t \in [-1, 1]^d$ and expect the model to answer $f(\mathbf{v}_t)$. The input is a vector of size $d + 2$, $\mathbf{x}_t = \text{Concat}(\mathbf{c}_{i_t} + \boldsymbol{\delta_t}, \tilde{y}_t, b_t)$, where:

- If $t \in [1, K]$, $\tilde{y}_t = f(\mathbf{v}_t) + \boldsymbol{\epsilon}, b_t = 1$, where $\boldsymbol{\epsilon}$ is a small gaussian noise with $\sigma = 0.1$.
- If $t \in [K + 1, K + 20]$, $\tilde{y}_t = 0, b_t = 0$.

The model should try to infer the mapping $f$ given the noisy observations during the first $K$ steps, then answer the queries during the next 20 steps. The exact form of $f$ decides the task difficulty; here, we consider two different settings:

1. $f$ is a randomly initialized linear layer with bias, $d = 12$.
2. $f$ is a two-layer MLP with tanh activation, the size of the hidden layer is 2, $d = 6$.

After $f$ is generated in each trial, we do an affine transformation $f \leftarrow af + b$ so that $f(\mathbf{v}_t)$ has zero mean and unit variance. We calculate the MSE loss over the whole trial as the criterion for model performance, i.e.,

$$L = \frac{1}{K + 20} \sum_{t=1}^{K+20} (f(\mathbf{v}_t) - y_t)^2.$$

The same loss is used for meta-training.

### A.4 SUPPLEMENTARY FIGURES

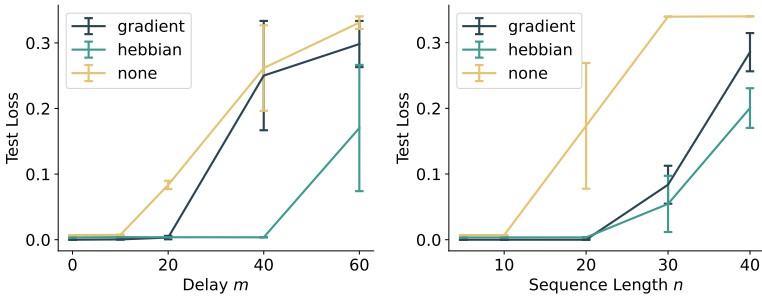

Figure 6: Performance of RNN models with different plasticity rules on the copying task. Error bars represent the SEM of four random runs. **Left:** Performance with different $m$ when $n = 5$. **Right:** Performance with different $n$ when $m = 0$.

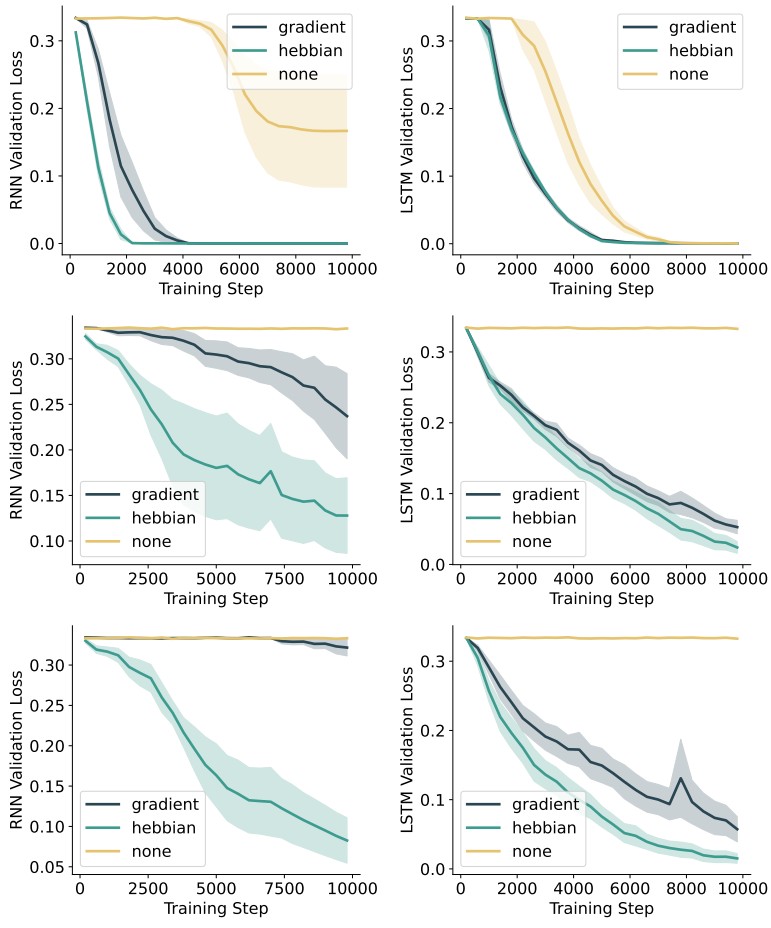

Figure 7: Validation loss curves for the copying task when $n = 20$. From top to bottom: $m = 0$, $m = 20$, $m = 40$. **Left:** RNN models. **Right:** LSTM models.

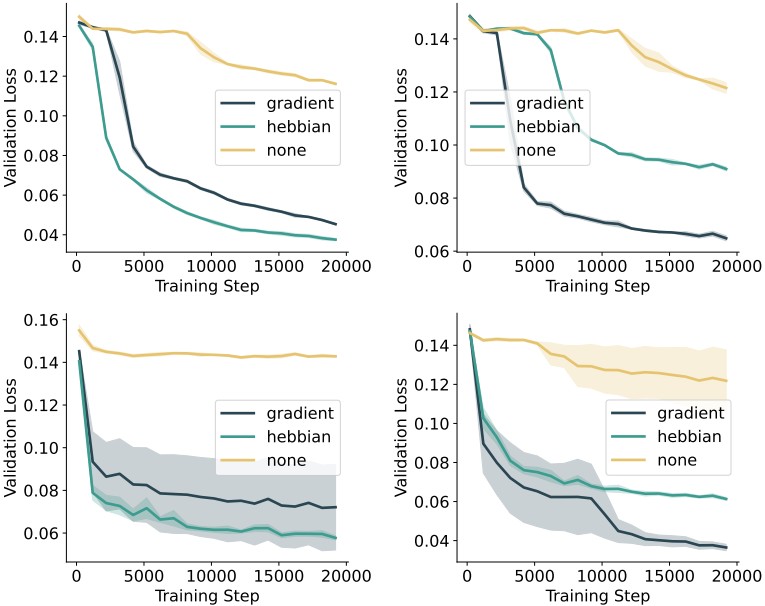

Figure 8: Results on the cue-reward association task when different learning rates are used. Note that the default learning rate is $10^{-3}$. **Top:** learning rate $= 3 \times 10^{-4}$. **Bottom:** learning rate $= 10^{-2}$. **Left:** RNN models. **Right:** LSTM models.

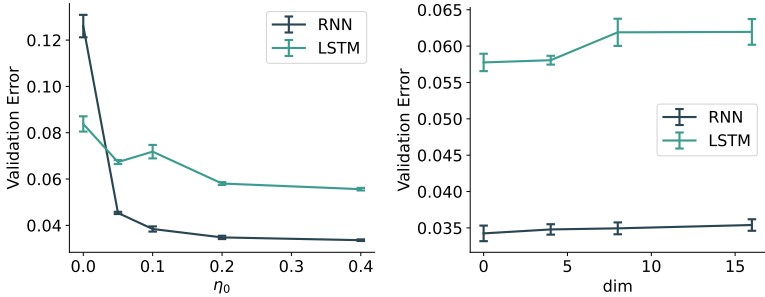

Figure 9: Ablation study on the cue-reward association task, Hebbian plasticity is used. **Left:** Effect of $\eta_0$ on final validation error. Note that $\eta_0 = 0$ means no plasticity. **Right:** Effect of $\dim(\tilde{\mathbf{y}}_t)$ on final validation error. For Hebbian plasticity, $\tilde{\mathbf{y}}_t$ has no significant effect on performance since it does not influence the model dynamics. It only has a minor effect on weight initialization (since the last linear layer is initialized in a fan-out manner).

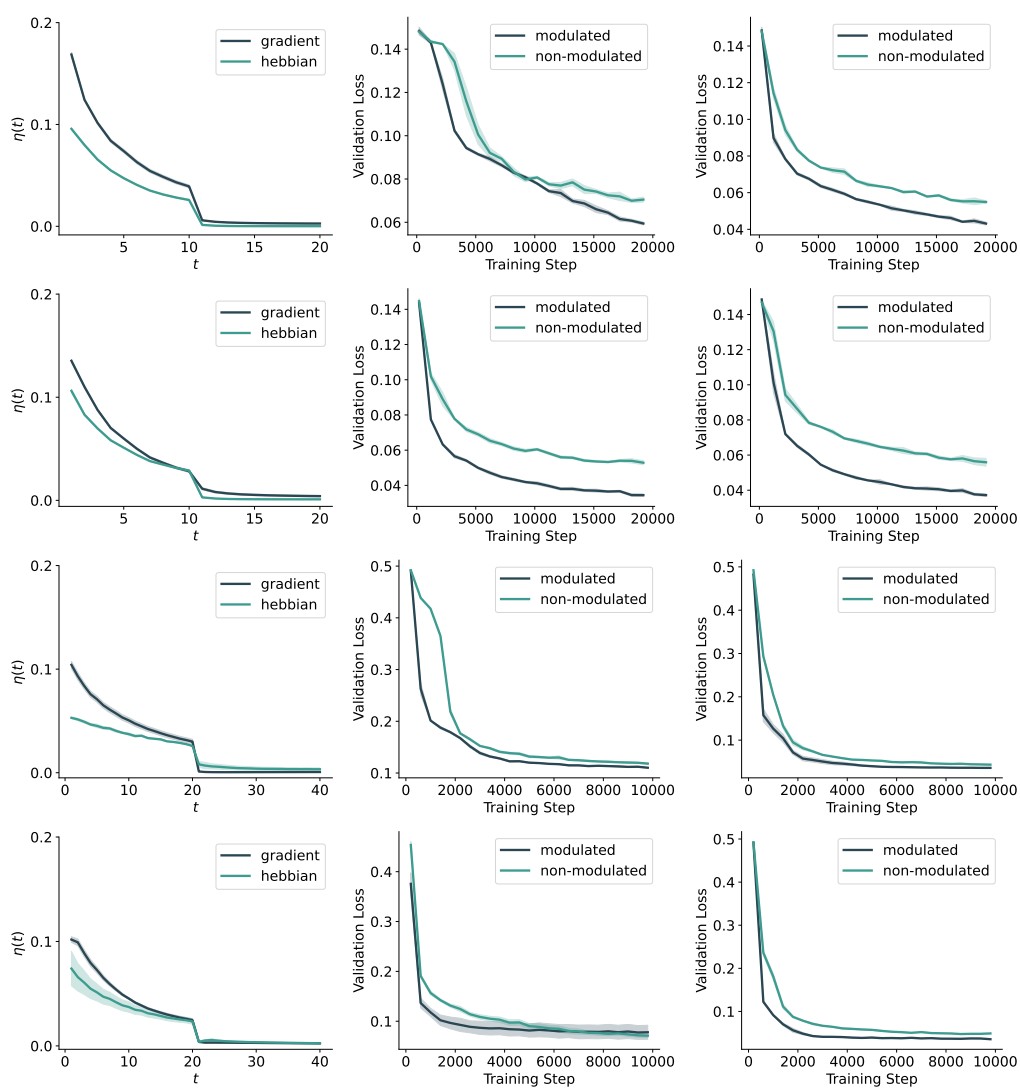

Figure 10: Trial-averaged curves of $\eta(t)$ (the left column) and the effect of neuromodulation on performance (the middle and the right column) in different tasks. The value of $\eta(t)$ is averaged over 6400 trials in the test set. The middle column shows results with Hebbian plasticity and the right column shows results with gradient-based plasticity. From top to bottom: 1) results of LSTM models on the cue-reward association task; 2) results of RNN models on the cue-reward association task; 3) results of LSTM models on the few-shot regression task, where the underlying mapping is an MLP, $K = 20$; 4) results of RNN models on the few-shot regression task.

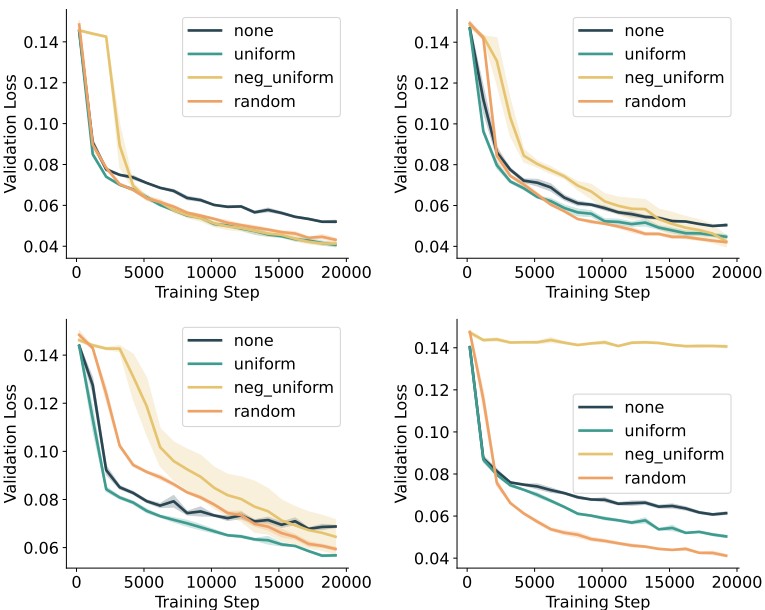

Figure 11: Comparison of different ways to initialize the learning rates $\alpha_{ij}$ on the cue-reward association task. 'none': no connection-specific learning rates. 'uniform': $\alpha_{ij} = 1$. 'neg_uniform': $\alpha_{ij} = -1$. 'random': $\alpha_{ij} \sim \mathcal{U}[-1, 1]$. **Top:** gradient-based plasticity. **Bottom:** Hebbian plasticity. **Left:** LSTM models. **Right:** RNN models.

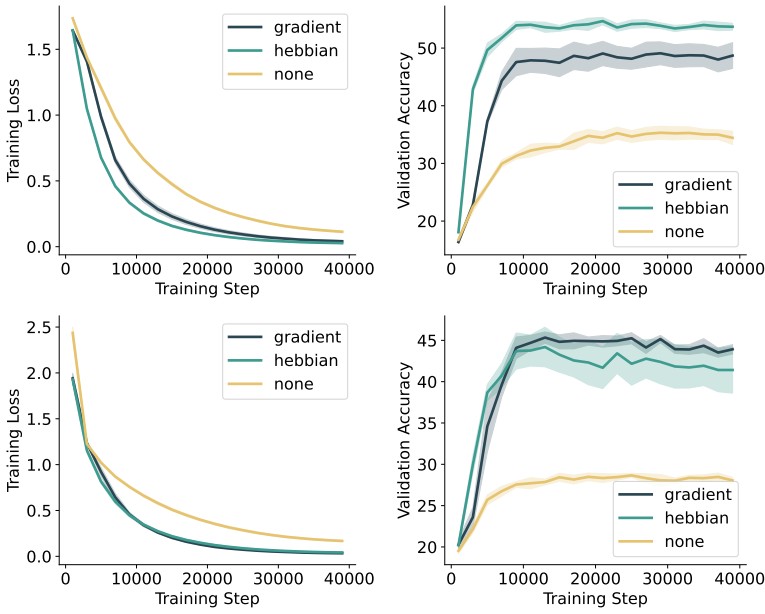

Figure 12: Training loss (left) and validation accuracy (right) curves of the one-shot image classification task on the CIFAR-FS (top) and miniImageNet (bottom) datasets. Here we use ResNet-12 as the encoder and the vanilla RNN (instead of LSTM) as the backbone of the recurrent module.

