# OpenReview forum: "Hebbian and Gradient-based Plasticity Enables Robust Memory and Rapid Learning in RNNs"
_ICLR.cc/2023/Conference — ICLR 2023 poster_

### Official Review · Reviewer_vDcd · 2022-10-15

**Confidence:** 3
**Correctness:** 3
**Technical Novelty And Significance:** 3
**Empirical Novelty And Significance:** 3
**Recommendation:** 6

**Clarity, Quality, Novelty And Reproducibility:**

Two main points require clarification:

- The description of the network operation is split over several section and somewhat  confusing. Most importantly, exactly which parameters are updated during the inner  vs. outer loop? This should be  specified explicitly.

- The outer loop meta-optimization process should be described more explicitly, especially for the gradient-based case. E.g. do we compute full second-order gradient-of-gradients? Or is there some first-order approximation (like first-order MAML, or REPTL) ?

For novelty, see  above.


**Strength And Weaknesses:**

Strengths:

- The paper tackles an interesting problem (learning to learn without necessarily having access to a supervised or even a reward  signal during each episode, i.e. the inner loop is unsupervised).

- The paper is interesting and may propose one novel method, i.e. network-generated gradient loss, though see below.

- The experiments, such as they are, offer reasonable evidence that the proposed methods are useful over the studied tasks


Weaknesses:

My main concern is about novelty. AFAICT the Hebbian portion of the method is identical to previous proposals that are cited in the paper.

Similarly, the gradient-based  method seems to be an example of meta-learning a loss function to perform gradient descent on in the inner loop. This approach is clearly not new, see e.g. https://arxiv.org/abs/1802.04821 , https://arxiv.org/abs/1906.05374 , https://arxiv.org/abs/2110.03909 and  the many references therein.  (Disclaimer: I am not affiliated with any of these authors)

Surprisingly, no previous paper on this approach (meta-learning a loss function) seems to be cited in the paper, unless I missed it.

There *may* be some novelty in the proposed approach, because here the synthetic loss  is an output of the learner network itself (as opposed to a separate dedicated network). IIUC this means that the synthetic loss itself is subject to learning and tuning during the inner loop (as  in Baik et al. mentioned above, but in a much more flexible manner). This also gives rise to interesting loops in the learning process (though the precise working of the algorithm is a bit unclear, see below).

If this is correct, and if this is the claim to novelty of the paper, it should be stated explicitly.

Additional possible  problems:
- The paper is somewhat unclear on certain points, which is important to better understand the proposed method. See below.
- The experiments are interesting, but perhaps a bit limited?
- Not really concerns, but suggestions for future work: the two approaches (Hebbian and gradient-based) seem fully compatible and it might be interesting to see what  happens when a network uses both together; the weight decay parameter, which is now just 1-eta, could include a separate learnable parameter as is Tyulmankov et al. and Rodriguez, Guo && Moraitis ICML 2022 (I am not affiliated with these authors).

**Summary Of The Paper:**

The paper compares two forms of meta-learning, with inner-loop updates based either on modulated Hebbian plasticity, or on gradient descent over a network-generated synthetic loss function.

Various experiments show that both forms of episodic plasticity (Hebbian and gradient-based) improve performance over using fixed recurrent networks, as in RL^2/L2RL. Network-controlled modulation of plasticity is also shown to be important.



**Summary Of The Review:**

== Update after reading the authors' response ==

I thank the authors for their clarifications and I have updated my review and score towards acceptance.

== Original review ==

Please cite more existing literature about the concept of meta-learned loss functions; explicitly state the novelty of the work; and clarify the working of the algorithm.

I am willing to increase my score depending on the author's response to the above, especially regarding novelty.

---

> ### Author Response · Authors · 2022-11-18
> **Author Response to Reviewer vDcd**
>
> Thank you for your valuable feedback! We attempt to address your concerns as follows:
>
> > Exactly which parameters are updated during the inner vs. outer loop?
>
> Only network weights $\mathbf{w}$ are updated in the inner loop. The initialization of network weights $\mathbf{\tilde w}$ and connection-specific learning rates $\mathbf{\alpha}$ are updated in the outer loop. For the gradient-based plasticity, the weight for calculating internal loss $\mathbf{w}_\text{out}$ is also updated in the outer loop.
>
> Although these have been mentioned in the original text, to make it more clear, **we added pseudocode on page 3 to elaborate the inner loop/outer loop structure**.
>
> > Do we compute full second-order gradient-of-gradients? Or is there some first-order approximation (like first-order MAML, or REPTL) ?
>
> The **full second-order gradient-of-gradients** is used, which makes the gradient-based plasticity slightly slower than Hebbian plasticity. For example, when training on the cue-reward association task:
>
> **Table R4.1**
>
> |                                        | Gradint-based | Hebbian | Non-plastic  |
> |----------------------------------------|---------------|---------|--------------|
> | Training time of 200 batches | 44s           | 36s     | 23s          |
>
>
> We left more details on the optimization algorithm in the appendix (Sec. A.2).
>
> > AFAICT the Hebbian portion of the method is identical to previous proposals that are cited in the paper.
>
> While the idea of neuromodulated Hebbian plasticity is not new, our work introduces critical updates. Please refer to the general response for more detailed discussions on this issue.
>
> > Similarly, the gradient-based method seems to be an example of meta-learning a loss function to perform gradient descent on in the inner loop. This approach is clearly not new. see e.g. https://arxiv.org/abs/1802.04821 , https://arxiv.org/abs/1906.05374 , https://arxiv.org/abs/2110.03909 and the many references therein.
>
> Indeed, the idea of meta-training the loss function has been applied in supervised learning and RL. However, **there are at least three aspects that separate our proposal of gradient-based plasticity from previous work**.
>
> To begin with, like what we discussed in Sec. 2, these methods still assume that **supervising signals (e.g., reward, labels) are explicitly made available to the base learner** (and the meta-learned network that generates the internal loss). In contrast, the inner loop in our method is unsupervised. As a result, **the task setting that these meta-learning methods aim to solve is less general**. Our work goes beyond the conventional definition of meta-learning tasks and can potentially be applied to any task with sequential input. For instance, it is hard to see how meta-learning the loss function can tackle the sequence copying task (Sec. 4.1).
>
> In addition, you mentioned that
> > There may be some novelty in the proposed approach, because here the synthetic loss is an output of the learner network itself (as opposed to a separate dedicated network). IIUC this means that the synthetic loss itself is subject to learning and tuning during the inner loop (as in Baik et al. mentioned above, but in a much more flexible manner). This also gives rise to interesting loops in the learning process.
>
> We agree with your understanding. In animal brains, neurons that encode the error or reward signals (e.g., dopaminergic neurons) receive input from brain regions that are subjected to neuromodulated plasticity themselves. Therefore, **having an internal loss that is more flexibly modulated by experience makes our method closer to the biological setting**.
>
> Finally, it is essential to note that in our method, **the update rule in the inner loop is much more flexible than gradient descent**. The connection-specific learning rates $\mathbf{\alpha}$ (which are randomly initialized and could be positive or negative) allow each synapse to implement its own learning dynamics. Our ablation study (Sec. 4.5) shows that using connection-specific learning rates consistently improves performance.
>
> Overall, we agree that meta-learning the loss function is an important and relevant line of work that we should mention in our paper. We thus **updated Sec. 2 to clarify the relationship between this line of work and our method**.

---

> > ### Comment · Reviewer_vDcd · 2022-11-18
> > **Response**
> >
> > Fair enough. I have updated my score towards acceptance, mostly on the strength of  the gradient-based contribution.
> >
> > Please include a mention of Miconi et al 2019 after Equation 2, since IIUC that's where the idea of placing plasticity under control of the network with eta(t) comes from.

---

> > > ### Author Response · Authors · 2022-11-19
> > > **Further Response**
> > >
> > > Thanks for your suggestions! We added a clarification after equation 2 (near the end of Sec. 3.2)
> > >
> > > > Previous work has shown the benefit of such adaptive learning rates on Hebbian plasticity \citep{miconi2018backpropamine}.
> > >
> > > We would like to point out that we have already acknowledged previous designs of global neuromodulation in the original manuscript (in Sec. 2). Still, we agree that adding further clarification is better!

---

### Official Review · Reviewer_fkhw · 2022-10-24

**Confidence:** 4
**Correctness:** 3
**Technical Novelty And Significance:** 4
**Empirical Novelty And Significance:** 3
**Recommendation:** 6

**Clarity, Quality, Novelty And Reproducibility:**

- the functional form of the internal loss was unclear (seems to aim to shrink the norm of the total output but it is unclear to me why that would be a generally useful task agnostic thing to do)
- the timescales at which various bits change in the model are hard to parse at times; some visual for the inner/outer loop, which parameters they change and based on which loss would be very useful to. navigate the model complexity
- the justification of the alternative models and the logic of their selection is critical for interpreting the results so it should be included in the main text
- the exact details of the experiments w.r.t. to losses and how things were set up is insufficient

**Strength And Weaknesses:**


+ Strength: impressive performance
+ S: analogies to biology as a motivation for the general approach
- Weakness: the logic of the optimization procedure is not spelled out as clearly as it should be
- W: alternative model comparison is restricted to variations of the same scheme, hard to evaluate the benefits of the approach more broadly w.r.t. alternative few shot learning approaches
- W: conceptually unclear to me if the hebbian vs gradient based distinction is a technical choice or if one of the two versions is championed in the paper


**Summary Of The Paper:**

The manuscript introduces a new way of learning task-specific unsupervised learning rule for RNNs in which the network computes not only its task output but also dynamically alters hyperparameters of its own learning process. The learning rule functional form is learned in a gradient-based outer loop based on task constraints provided via a validation set. The model is validated on a wide range of tasks and shows interesting few shot learning performance improvements relative to a static network.


**Summary Of The Review:**

Overall an intriguing idea and interesting results, but the presentation of the core methodology and its justification is too sparse and needs substantial improvement before publication.

---

> ### Author Response · Authors · 2022-11-18
> **Author Response to Reviewer fkhw**
>
> Thanks for clearly pointing out the strengths and weaknesses of our paper. We attempt to address your concerns as follows:
>
> > the logic of the optimization procedure is not spelled out as clearly as it should be; the timescales at which various bits change in the model are hard to parse at times
>
> To clarify the optimization procedure, **we added pseudocode on page 3 to elaborate the inner/outer loop structure**.
>
> > conceptually unclear to me if the Hebbian vs gradient based distinction is a technical choice or if one of the two versions is championed in the paper
>
> We want to point out that **both rules have pros and cons**. Please refer to our general response for additional clarifications.
>
> > alternative model comparison is restricted to variations of the same scheme, hard to evaluate the benefits of the approach more broadly w.r.t. alternative few-shot learning approaches
>
> We agree that more comparisons help evaluate our methods. Among the tasks we considered, the one-shot image classification task is a standard benchmark for meta-learning. **We were thus able to compare our performance with other few-shot learning methods (shown in Table 1)**. Since the other three tasks are designed by ourselves and do not exist in prior work, comparing our approach with previous ones on these tasks takes more effort.
>
> On the one-shot image classification task, **our plastic RNNs have comparable performance to other meta-learning methods (e.g., MAML, ProtoNet) when we limit the visual encoder to be a 4-layer CNN**. However, unlike the SOTA methods, we did not find plastic networks significantly benefit from a deep vision encoder. We note that **methods that get the best results on few-shot learning benchmarks are also exclusively designed for few-shot image classification**. Unlike our plastic RNNs, these methods are difficult to apply to other learning problems or memory tasks. Instead of striving to get the best performance on a specific task, our goal is to build a general architecture that tackles a wide range of memory and learning problems.
>
> > the justification of the alternative models and the logic of their selection is critical for interpreting the results so it should be included in the main text
>
> Our main results compare static RNNs and RNNs with Hebbian or gradient-based plasticity with the same number of parameters. These comparisons clearly illustrate the benefit of two different forms of plasticity.
>
> We previously showed the results of various alternative models in our ablation study in the appendix. To better present the ablation results, we have moved some of them to the main text (Figure 5).
>
> > the functional form of the internal loss was unclear (seems to aim to shrink the norm of the total output but it is unclear to me why that would be a generally useful task-agnostic thing to do)
>
> The internal loss is basically the L2 norm of the model output. However, **our update rule does not necessarily shrink this norm** since the connection-specific learning rates $\mathbf{\alpha}$ could be positive or negative. **We hypothesize that such design gives the model sufficient freedom to evolve customized learning rules according to the task setting.** This hypothesis is validated in our diverse memory and learning tasks.
>
> The main reason for using the L2 norm (instead of other loss functions) is that it makes the dynamics of the last linear layer Hebbian (see Sec. A.1.), which strengthens the relationship between the two proposed plasticity rules. Nevertheless, exploring other forms of internal loss would be an interesting future direction.
>
> > the exact details of the experiments w.r.t. to losses and how things were set up is insufficient
>
> We have now included more details on the task setup in the appendix (Sec. A.3). For full reproducibility, we plan to release our code if our paper gets accepted.

---

> > ### Comment · Reviewer_fkhw · 2022-11-18
> > **Response**
> >
> > Thanks for the clarifications. I still believe that the loss function is ad hoc and could have been justified better but the addition of the pseudocode helps with reproducibility if not understanding. It was useful to see the expanded across model comparison. Overall, I believe my original score remains an accurate reflection of my evaluation of the paper, so I'll maintain my score of 6.

---

### Official Review · Reviewer_5CwK · 2022-10-24

**Confidence:** 3
**Correctness:** 2
**Technical Novelty And Significance:** 3
**Empirical Novelty And Significance:** 2
**Recommendation:** 6

**Clarity, Quality, Novelty And Reproducibility:**

The proposed method requires more clarification for the meta-learning update rule. The proposed method is not very novel, and the quality of the paper is negatively affected by the lack of more thorough comparisons.

**Strength And Weaknesses:**

The proposed meta-learning approach is interesting especially from a neuroscience perspective (though not novel, e.g. Najarro and Risi, NeurIPS, 2020). However, I’m not convinced that the proposed model has any advantages compared to the previously proposed methods. In particular, almost all the comparisons are between the proposed method and non-plastic RNNs which I found insufficient.

$\textbf{Questions}$:

1- Almost all the comparisons are with what is called non-plastic RNNs in the paper. The motivation behind using non-plastic RNNs as the baseline model for comparisons is not explained in the paper. Specifically, the initialization of weights could place the untrained RNNs in a disadvantaged dynamical regime compared to the trained RNNs. It is hard to imagine a situation where non-plastic neural networks are considered as an alternative for how brains function, except in cases where, for example, the RNNs are initialized at the edge of chaos (criticality) which could give them specific advantages (better short-term memory, etc.) Can the authors elaborate more on the motivation behind comparing only with untrained RNNs?

2- The update rules of the outer (meta-learning) rule are not shown in the paper. It was only mentioned that gradient descent was used for the outer loop, but the loss function and the target variables are not explained. Was the MSE loss shown in figures 2-3 used for the outer loop updates? Also, mentioned in section 3.2 “ [...] network parameters, including those that define the learning rules in the inner loop, are meta-trained with gradient descent.” A list of parameters meta-trained with gradient descent would be helpful.

3- Related to comment # 1, it would be interesting to see comparisons with RNNs trained in a supervised (or unsupervised) fashion (no meta-learning). The main advantages of the meta-learning approach in associative memory and few-shot learning would be more clear with those comparisons.

4- One general comment: Although a key concept in the paper, the connection with meta-learning did not become clear until the end of the introduction (not even mentioned in the abstract). Concepts like “self-determined target” and “neuromodulated plasticity” are used a few times throughout the introduction before mentioning that the proposed method is a meta-learning algorithm. Because of that, It's difficult to follow the introduction and understand the main question of the paper. I suggest that the authors rephrase the introduction and highlight the connections with meta-learning at the very beginning.


**Summary Of The Paper:**

Inspired by the evolved role of plasticity in neural circuits, this paper proposes a meta-learning approach for training RNNs. The method consists of an inner loop with a learning rule that updates the parameters of an RNN, and an outer loop that learns a function for the update rule.

**Summary Of The Review:**

Although the proposed meta-learning algorithm is interesting (not novel), the comparisons with untrained (or non-plastic) RNNs is not sufficient for demonstrating the strengths of the proposed method.

---

> ### Author Response · Authors · 2022-11-18
> **Author Response to Reviewer 5CwK**
>
> Thanks for your valuable feedback! We feel that there might be some misunderstanding of our method and experiments. We attempt to address your concerns as follows.
>
> > A list of parameters meta-trained with gradient descent would be helpful.
>
> Both the initialization of network weights $\mathbf{\tilde w}$ and connection-specific learning rates $\mathbf{\alpha}$ are meta-trained with GD. For the gradient-based plasticity, we also update the weight for calculating internal loss $\mathbf{w}_\text{out}$ in the outer loop. **We now add pseudocode on page 3 to elaborate the inner loop/outer loop structure**.
>
> > It was only mentioned that gradient descent was used for the outer loop, but the loss function and the target variables are not explained. Was the MSE loss shown in figures 2-3 used for the outer loop updates?
>
> Yes, the loss used for meta-training is the same as the loss shown in our figures. For better clarity, **we have added more task details in the appendix (Sec. A.3).**
>
> > Almost all the comparisons are with what is called non-plastic RNNs in the paper. ... Specifically, the initialization of weights could place the untrained RNNs in a disadvantaged dynamical regime compared to the trained RNNs ... Can the authors elaborate more on the motivation behind comparing only with untrained RNNs?
>
> As baseline models, **the weights $\mathbf{\tilde w}$ of non-plastic RNNs are meta-trained in the same way as plastic RNNs**. These models are certainly not untrained. The only difference between non-plastic and plastic RNNs is that non-plastic RNNs do not have a plastic component $\mathbf{w}$ in their weights, i.e., their weights remain fixed within a trial.
>
> In the paper you mentioned (Meta-Learning through Hebbian Plasticity in Random Networks, Najarro and Risi, NeurIPS, 2020), the setting is very different from what we do here. In that paper, **the networks start with random initializations for each trial**, and evolutionary algorithms are used as the meta-training algorithm. In our paper, the networks **begin with an initialization that is meta-learned with GD**.
>
> > It would be interesting to see comparisons with RNNs trained in a supervised (or unsupervised) fashion (no meta-learning). The main advantages of the meta-learning approach in associative memory and few-shot learning would be more clear with those comparisons.
>
> If we understand correctly, you propose that we could directly train a network in each trial as a baseline method. In this case, since the network is trained on a minimal number of samples, we would expect them to perform poorly compared to meta-learning methods such as plastic RNNs.
>
> This is indeed the case. The MAML paper (Model-Agnostic Meta-Learning for Fast Adaptation of Deep Networks, 2017) showed (in Table 1) that directly training models on the small training set only achieves $28.9 \pm 0.5 \%$ accuracy on the 1-shot 5-way image classification task of miniImageNet Dataset. In our experiments, **all models significantly outperform this baseline** (See the second column of Table 1 in our paper). For example, even non-plastic RNNs achieve $44.1 \pm 0.7\%$ accuracy in our experiments. Since this baseline is very weak, it is usually omitted in recent work on meta-learning.
>
> > The proposed method is not very novel, and the quality of the paper is negatively affected by the lack of more thorough comparisons.
>
> While the Hebbian plasticity part is partly based on previous work, we believe the proposed gradient-based plasticity is quite novel in meta-learning. Please refer to our general response for more thorough discussions on novelty.
>
> We were wondering if our previous clarification resolves your concern about the lack of thorough comparison. To the best of our knowledge, **prior works on Hebbian plasticity are all limited to comparing networks with one type of plasticity (and its variants) with networks with no plasticity**. In contrast, we additionally compared two qualitatively different types (local and non-local) of plasticity rules.

---

> > ### Comment · Reviewer_5CwK · 2022-11-21
> > **Thanks for the clarifications**
> >
> > Thanks for the responses and the clarifications. The addition of the pseudocode and the general explanations help a lot. As a result, I'll improve my score.

---

### Official Review · Reviewer_x3Sh · 2022-10-25

**Confidence:** 4
**Correctness:** 4
**Technical Novelty And Significance:** 2
**Empirical Novelty And Significance:** 2
**Recommendation:** 6

**Clarity, Quality, Novelty And Reproducibility:**

The text is well-written; the descriptions of the experiments contain sufficient information for reproducibility. The figures are good at conveying the results of the experiments (perhaps, they could be moved to the main text; also, little titles for individual panels within the figures may help parse them faster).

**Strength And Weaknesses:**

This is a thoroughly-designed and well-described study of plasticity in recurrent neural networks, which builds upon previous results by Jeff Clune, Ken Stanley, and colleagues, known to the ICLR community.

The paper introduces several interesting updates on top of the previous works. First, the “learning rate of plasticity” \eta here is time-dependent, which supposedly allows the models to turn on the synaptic plasticity during one/few-shot learning, and to turn it off to retain the learned knowledge during testing. Second, in the gradient-based method, the Authors introduce internal loss. This loss relies on an extra output of the RNN, conditioned upon the RNN’s weights trained in the outer loop, and is used to guide the network’s plasticity in the inner loop. For both these updates, the Authors provide (in the Appendix) the ablation studies showing how different values of these parameters affect the models’ performance.

The internal loss part seems especially interesting because it may link this work to a biological phenomenon of motivation. Previously. motivations in machine learning were set to reflect observable quantities such as curiosity; conversely, the internal cost functions here are defined by the model itself to reflect the important task-related quantities.

Another strength of this work lies in the comprehensive testing of the models which includes standard benchmarks and thorough ablation studies. It would be nice though to move more figures to the main text, if possible, as they show important results.

Speaking of weaknesses, the Hebbian-plasticity part of the work is highly similar to that in a series of works by Miconi et al, mentioned by the Authors (the similarity is also acknowledged by the Authors). The differences (which need to be clearly summarized somewhere in the text!) seem to be limited to 1) the time-dependent internal learning rate for synaptic plasticity, 2) not using Oja’s rule, and 3) using a (more complex) image classification task instead of the Omniglot. Furthermore, the proposed gradient-based rule seems to be only efficient in the few-shot regression task among the used benchmarks. These two factors limit the contribution of this work to the field. I also found the biological insights a bit preliminary – though they do not constitute the main result of the work.


**Summary Of The Paper:**

In this paper, the Authors design and implement recurrent neural networks with two types of plasticity: classical Hebbian and gradient-based w.r.t. an internal loss function. The Authors then test their models on a set of memory/learning tasks including copying, cue-reward association, image classification, and regression. They find that the new gradient-based approach fares better on the regression task while the classical Hebbian approach is better in copying and image classification tasks. An ablation study is provided to assess the importance of the models’ parameters.

**Summary Of The Review:**

This is a comprehensive study describing two mechanisms of synaptic plasticity in RNNs. The properties of the described models are thoroughly tested. In the future, some of the design choices made here (quite interesting!) may help build neuroscience-relevant models. The drawback is in high similarity to prior work (Hebb’s rule part) and high similarity to Hebb’s rule scores on benchmarks (gradient part).

---

> ### Author Response · Authors · 2022-11-18
> **Author Response to Reviewer x3Sh**
>
> Thanks for your thorough review! We especially appreciate your interest in our idea and insightful suggestions that help us make the paper better. Following your recommendation, we have moved some figures to the main text to present the ablation results better. Please refer to our general response for detailed discussions on novelty.

---

> > ### Comment · Reviewer_x3Sh · 2022-11-18
> > **Re: Author response**
> >
> > Thanks for your response and clarifications, including an explanation and an ablation experiment highlighting the role of the internal learning rate in preventing the plastic parameters W from abrupt changes. While this is an interesting change (on top of the other strengths of the paper), the similarity to the past work remains both architecturally and conceptually (i.e. a similar general structure + having a (novel) tweak to stabilize training), therefore I maintain my score.
> >
> > Provided that the role of the internal learning rate in stable training is one of this paper's highlights, I think it would be interesting to see how it compares to previously used approaches mentioned by the Authors (i.e. Oja's rule, weight clipping, etc.) but I understand that - unless I'm missing something here - such comparisons are currently impossible due to the time constraints.

---

> > > ### Author Response · Authors · 2022-12-02
> > > **Further Response**
> > >
> > > Thanks for your feedback! Although we feel our technical updates on stabilizing the training of plastic RNNs could be beneficial for future research in this direction, we want to clarify that this is not our main contribution (which is why we did not include comparisons between different ways of implementing the Hebbian rule). Instead, we believe our proposal of gradient-based plasticity and comparing these two plasticity rules on a new set of memory and learning tasks constitute the paper's main contribution.

---

### Public Comment · ~Timoleon_Moraitis1 · 2022-11-17
**Missing reference to closely related work**

Dear authors,

We would like to draw your attention to the latest work that also uses meta-learned Hebbian plasticity in recurrent neural networks [1] but is not cited in your manuscript.

[1] extends Hebbian plasticity with short-term dynamics, applies it to similar problems as the study reviewed here, outperforms several baselines including plasticity of the Miconi type that the authors used, and also extends its applicability to reinforcement learning problems that the authors here characterize as challenging and left for future study.

We believe that this reference is highly relevant and should be cited in the present manuscript for an accurate depiction of the current state of the field.

Sincerely,

Timos Moraitis

[1] Rodriguez et al., *Short-Term Plasticity Neurons Learning to Learn and Forget*, ICML 2022 https://proceedings.mlr.press/v162/rodriguez22b.html

---

> ### Comment · Reviewer_x3Sh · 2022-11-17
> **Re: reference**
>
> Thanks for the reference. I would like to point out though that it does not deem reasonable to expect the Authors to have included this reference in the original submission as the ICML'22 conference took place in late July, likely at the same time when this early (judging by the number) submission was being written up.

---

> ### Author Response · Authors · 2022-11-18
> **Re: Missing reference to closely related work**
>
> Thanks for your timely feedback! We agree that [1] is an important extension of the previous methods. We added references to the paper in both Sec. 1 and Sec. 2. we also modified the discussion section in light of these results.
>
> However, if we understand correctly, [1] considers a very different set of problems in its experiments compared to our work. [1] introduces impressive progress on RL tasks but doesn't consider sequential memory tasks and few-shot learning tasks. The cue-reward association task is similar to the associative recall task in [1] but has a different setting.
>
> **Although both are based on [2], our work and [1] contribute to the field in different ways.** Instead of investigating the advantages of short-term dynamics as in [1] (mostly in RL tasks), we are primarily interested in how RNNs can benefit from local and non-local plasticity rules (in memory and few-shot learning tasks).
>
> 1. Rodriguez, Hector Garcia, Qinghai Guo, and Timoleon Moraitis. "Short-Term Plasticity Neurons Learning to Learn and Forget." International Conference on Machine Learning. PMLR, 2022.
> 2.  Thomas Miconi, Aditya Rawal, Jeff Clune, and Kenneth O. Stanley. Backpropamine: training self-modifying neural networks with differentiable neuromodulated plasticity. In International
> Conference on Learning Representations, 2019.

---

### Author Response · Authors · 2022-11-18
**General Response**

We thank all reviewers and community members for their efforts in evaluating the paper and writing suggestions that help us improve our work. Here we respond to some common concerns raised by reviewers. We modified the manuscript accordingly.

### Clarity of our method

Although reviewer x3Sh commented that our manuscript is "well-written", other reviewers felt that our method was not sufficiently explained. To clarify the inner-loop/outer-loop structure in our approach, including how to update parameters, **we added pseudocode on page 3 to summarize our algorithm. We recommend you take a look if you are previously confused**.

### Novelty of Gradient-based Plasticity

To the best of our knowledge, **the proposed gradient-based plasticity is unique in the field of meta-learning**. Although there are several relevant lines of work such as writing memories with GD [4] and meta-learning a loss function (thanks reviewer vDcd for pointing this out), our **unsupervised problem setting** (the inner loop is unsupervised) and **flexibility of the inner loop learning rule** (using self-generated target and connection-specific learning rates $\mathbf{\alpha}$) separate our method from prior ones. For more detailed discussions, please refer to our response to reviewer vDcd.

### Differences from Previous work on Hebbian Plasticity

We would like to clarify the contribution of Hebbian plasticity part of our work since reviewer x3Sh, 5CwK, and vDcd all raised related questions. The idea of applying Hebbian plasticity into RNNs is not new, and we acknowledged similar designs in our paper. We are mostly inspired by the work of Miconi et al. [1, 2]. However, **our paper introduces critical updates that we summarize as the following three major points.**

First, prior work noted that plastic RNNs are unstable and proposed using Oja's rule [1], clipping the weights to a small range [2], or normalizing the plastic components [6] to stabilize training. **Unlike previous work, we prevent the plastic parameters $\mathbf{w}$ from changing too quickly by tuning down the internal learning rate** when $\Delta \mathbf{w}$ is large (see equation 2, where we use $\delta_t$ to denote the change of weights). This strategy works well on the tasks we considered, and **we added an experiment in the ablation study (Sec. 4.5) to illustrate the necessity of doing so**. In short, we found that **without this design, RNN models with Hebbian plasticity explode (training and test loss become nan) in 3 out of 4 random runs on the copying task**. Apart from that, we do gradient clipping during meta-training and use a decay term in the plasticity rule to further make network weights less prone to exploding.

In addition, inspired by the current understanding of synaptic plasticity in animals (first paragraph of Sec. 1), we are more focused on the role of plasticity in memory and few-shot learning. **By testing models on a range of tasks not seen in the same line of work on Hebbian Plasticity**, we believe **our empirical results significantly contribute to understanding plasticity's benefits in RNNs**. Although the one-shot classification task (Sec. 4.3) is seen in [1], using more challenging image datasets (miniImageNet and CIFAR, instead of Omniglot used in [1]) shows more substantial proof of the effectiveness of our method. Using more standard datasets also allows us to better evaluate the performance w.r.t. other recent few-shot learning methods. Moreover, while other associative memory tasks exist in prior work [1, 3], our cue-reward association task (Sec. 4.2) has a different setting.

Lastly, as acknowledged by reviewer x3Sh, **we conduct thorough and systematic analysis on various aspects of our method**, most of which are absent from previous papers. These include (1) showing how performance changes with maximal internal learning rates, (2) investigating the effect of connection-specific learning rates $\mathbf{\alpha}$, including different ways to initialize it, (3) the benefit of the time-varying global learning rate $\eta(t)$ in different tasks, (4) doing experiments on both RNN and LSTM and compare their differences, and (5) examining other hyperparameters such as the meta learning-rate. **We put many relevant figures in the appendix (Sec. A.4), but we have moved some of them to the main text (in Figure 5)** (Thanks to the suggestion of x3Sh). We believe these technical explorations are critical, considering that plastic RNNs are potential modeling tools for neuroscientists.

---

> ### Author Response · Authors · 2022-11-18
> **General Response (Part 2)**
>
> ### Comparing two plasticity rules
>
> Although our motivation for proposing the gradient-based plasticity rule is to address the limitation of classical Hebbian plasticity, we don't intend to argue that one plasticity rule is better than the other. Instead, **just like different plasticity rules have been found in different brain regions accountable for different cognitive functions [5], we also believe that different plasticity rules are suitable for different tasks in ANNs.** We modified part of the Discussion (Sec. 5) to clarify our message.
>
> ### Reproducibility
>
> While reviewer x3Sh found our description "contains sufficient information for reproducibility", reviewer vDcd and fkhw raised concerns. In the updated manuscript, **we include comprehensive explanations of experiments in the appendix (Sec. A.2, A.3)**. For full reproducibility, **we also plan to release our code if the paper gets accepted**.
>
> ### Other changes to the manuscript
>
> * We updated the discussion part on biological insights and future work.
> * We fixed an issue that causes the error bars in some plots to be slightly smaller. The figures are thus updated.
> * We moved many task details to the appendix (Sec. A.3) to fit the paper into nine pages.
> * Minor changes in wording or phrasing.
>
> ### References
>
> 1. Thomas Miconi, Kenneth Stanley, and Jeff Clune. Differentiable plasticity: training plastic neural networks with backpropagation. In International Conference on Machine Learning, pp. 3559–3568. PMLR, 2018.
> 2. Thomas Miconi, Aditya Rawal, Jeff Clune, and Kenneth O. Stanley. Backpropamine: training self-modifying neural networks with differentiable neuromodulated plasticity. In International
> Conference on Learning Representations, 2019.
> 3. Jimmy Ba, Geoffrey E Hinton, Volodymyr Mnih, Joel Z Leibo, and Catalin Ionescu. Using fast weights to attend to the recent past. Advances in neural information processing systems, 29, 2016.
> 4. Bartunov, Sergey, et al. Sergey Bartunov, Jack Rae, Simon Osindero, and Timothy Lillicrap. Meta-learning deep energy-based memory models. In International Conference on Learning Representations, 2020.
> 5. Jeffrey C. Magee and Christine Grienberger. Synaptic plasticity forms and functions. Annual Review of Neuroscience, 43(1):95–117, 2020. doi: 10.1146/annurev-neuro-090919-022842
> 6. Rodriguez, Hector Garcia, Qinghai Guo, and Timoleon Moraitis. "Short-Term Plasticity Neurons Learning to Learn and Forget." International Conference on Machine Learning. PMLR, 2022.

---

### Author Response · Authors · 2022-12-02
**Thanks again for reviewers' feedback**

We thank all reviewers for their insightful reviews and prompt feedback after our response. These valuable, constructive suggestions gave us immense help in organizing a clear and rigorous paper. We are glad to hear that our revision of the manuscript successfully addressed some major concerns of reviewers, including clarity and reproducibility (reviewer 5CwK and fkhw), novelty, and relationship to prior work (reviewer vDcd).

After reading the reviews, we became more confident that our proposal of gradient-based plasticity and the comparison of two plasticity rules on a range of new tasks could be interesting to both ML and neuroscience communities. In addition to what we discussed in the paper, we are excited that hear that reviewers also proposed some interesting future directions that we previously have not thought of, including combining local and non-local plasticity rules (reviewer vDcd) and modeling the biological phenomenon of motivation with the gradient-based plasticity (reviewer x3Sh).

---

### Decision · Program_Chairs · 2023-01-20

**Decision:**

Accept: poster

**Justification For Why Not Higher Score:**

There was clear agreement amongst the reviewers that this is a relatively incremental contribution, which is also my assessment as an AC. Thus, although it is appropriate for acceptance, it is not sufficiently novel or cutting-edge to warrant a spotlight or oral.

**Justification For Why Not Lower Score:**

Given that the paper is clear, most of the concerns were well addressed in the rebuttal, and that much of the concern was about novelty in a manner that was addressed in part by better positioning relative to past work, it does not seem appropriate to reject this paper.

**Metareview: Summary, Strengths And Weaknesses:**

This paper presents an approach to training recurrent neural networks that combines Hebbian weight updates with gradient descent. The authors show that this approach improves few-shot learning in memory and association based tasks.

The strengths of this paper are its clarity, and post rebuttal, its reproducibility, both of which will allow it to serve as the basis for future work.

The principal weakness is that the novelty is somewhat limited.

The reviewers all noted the issue of novelty, and originally, several were concerned about reproducibility. However, following the authors' replies, the reviewers agreed that with sufficient references to past work and clarification the novel contributions are sufficiently clear to warrant acceptance and scores were bumped up. The final score was still technically borderline (average score of 6), but there was a total consensus on this score after rebuttal (all reviewers gave a 6), which indicates sufficient agreement that this paper is appropriate for acceptance.

**Note From Pc:**

if the above contains the word "oral" or "spotlight" please see: "oral" presentation means -> notable-top-5% and "spotlight" means -> notable-top-25%. As stated in our emails, we are disassociating presentation type from AC recommendations

**Summary Of Ac-Reviewer Meeting:**

This was technically a borderline paper based on the average score, but my assessment as an AC was that it is not actually borderline, because the reviewers are in quite clear agreement that the paper is clear and makes some decent contributions. No reviewer felt that this paper should be rejected, and so, I do not believe this represents a borderline case.